# Herpes Simplex Virus Infection Alters the Immunological Properties of Adipose-Tissue-Derived Mesenchymal-Stem Cells

**DOI:** 10.3390/ijms241511989

**Published:** 2023-07-26

**Authors:** Anikó Kun-Varga, Barbara Gubán, Vanda Miklós, Shahram Parvaneh, Melinda Guba, Diána Szűcs, Tamás Monostori, János Varga, Ákos Varga, Zsolt Rázga, Zsuzsanna Bata-Csörgő, Lajos Kemény, Klára Megyeri, Zoltán Veréb

**Affiliations:** 1Regenerative Medicine and Cellular Pharmacology Laboratory, Department of Dermatology and Allergology, University of Szeged, H-6720 Szeged, Hungary; kun-varga.aniko@med.u-szeged.hu (A.K.-V.); gubanbarbi@gmail.com (B.G.); guba.melinda@med.u-szeged.hu (M.G.); szucs.diana@med.u-szeged.hu (D.S.); monostori.tamas.bence@med.u-szeged.hu (T.M.); kemeny.lajos@med.u-szeged.hu (L.K.); 2Doctoral School of Clinical Medicine, University of Szeged, H-6720 Szeged, Hungary; 3Biobank, University of Szeged, H-6720 Szeged, Hungary; miklos.vanda96@gmail.com; 4HCEMM-SZTE Skin Research Group, University of Szeged, H-6720 Szeged, Hungary; shparvaneh79@gmail.com (S.P.); bata.zsuzsa@med.u-szeged.hu (Z.B.-C.); 5Interdisciplinary Research Development and Innovation Center of Excellence, University of Szeged, H-6720 Szeged, Hungary; 6Dermatosurgery and Plastic Surgery Unit, Department of Dermatology and Allergology, University of Szeged, H-6720 Szeged, Hungary; varga.janos@med.u-szeged.hu (J.V.); varga.akos@med.u-szeged.hu (Á.V.); 7Department of Pathology, University of Szeged, H-6720 Szeged, Hungary; razga.zsolt@med.u-szeged.hu; 8Department of Medical Microbiology, University of Szeged, H-6720 Szeged, Hungary; megyeri.klara@med.u-szeged.hu

**Keywords:** adipose-derived mesenchymal stem cells, inflammation, gene expression

## Abstract

The proper functioning of mesenchymal stem cells (MSCs) is of paramount importance for the homeostasis of the body. Inflammation and infection can alter the function of MSCs, which can also affect the regenerative potential and immunological status of tissues. It is not known whether human herpes simplex viruses 1 and 2 (HSV1 and HSV2), well-known human pathogens that can cause lifelong infections, can induce changes in MSCs. In non-healing ulcers, HSV infection is known to affect deeper tissue layers. In addition, HSV infection can recur after initially successful cell therapies. Our aim was to study the response of adipose-derived MSCs (ADMSCs) to HSV infection in vitro. After confirming the phenotype and differentiation capacity of the isolated cells, we infected the cells in vitro with HSV1-KOS, HSV1-532 and HSV2 virus strains. Twenty-four hours after infection, we examined the gene expression of the cells via RNA-seq and RT-PCR; detected secreted cytokines via protein array; and determined autophagy via Western blot, transmission electron microscopy (TEM) and fluorescence microscopy. Infection with different HSV strains resulted in different gene-expression patterns. In addition to the activation of pathways characteristic of viral infections, distinct non-immunological pathways (autophagy, tissue regeneration and differentiation) were also activated according to analyses with QIAGEN Ingenuity Pathway Analysis, Kyoto Encyclopedia of Genes and Genome and Genome Ontology Enrichment. Viral infections increased autophagy, as confirmed via TEM image analysis, and also increased levels of the microtubule-associated protein light chain 3 (LC3B) II protein. We identified significantly altered accumulation for 16 cytokines involved in tissue regeneration and inflammation. Our studies demonstrated that HSV infection can alter the viability and immunological status of ADMSCs, which may have implications for ADMSC-based cell therapies. Alterations in autophagy can affect numerous processes in MSCs, including the inhibition of tissue regeneration as well as pathological differentiation.

## 1. Introduction

Controlling the function and fate of stem cells is crucial for the homeostasis of the body. Changes, such as inflammation and infection, can alter the biological status of stem cells and, in particular, the regeneration potential of the tissue and its immunological status [1]. Our interest in this study was to examine the effects of human herpes simplex viruses 1 and 2 (HSV-1 and HSV-2) on mesenchymal stem cells (MSCs).

Herpesviruses are human pathogens causing life-long infections with clinical manifestations ranging from mucosal eruptions to disseminated infections that can affect visceral organs and the central nervous system [2]. Initially, the viruses only infect epithelial cells, but eventually, they spread to innervating nerves and ganglia. Subsequently, the virus remains latent in neurons, and the reactivation of viral action inflicts painful ulcerative lesions of the oropharynx, skin and mucosal tissue in oral and genital herpes. Most HSV-1 and HSV-2 lesions tend to occur during reactivation. Although lesions occur less frequently during primary infection, lesions occurring during primary HSV infection are well documented, especially among younger patients [3]. Moreover, latent HSV infection can be associated with several other diseases, such as atherosclerosis, multiple sclerosis, Alzheimer’s disease and irritable bowel syndromes [4,5,6].

HSV reactivation is common in transplant patients and is the second most common cause of viral infection occurring after transplantation [7,8]. Changes in transplantation techniques in recent decades have resulted in deeper and longer-lasting immunodeficiency. As a result, the incidence of HSV infections resistant to antiviral agents, such as acyclovir, has increased [3,9,10,11]. Primary infection from transplanted tissues is uncommon but has been observed in renal transplant recipients [12,13]. A high percentage of HSV infection in transplantation cases is attributable to reactivation, which is undoubtedly due to the high seroprevalence in the population. Seroprevalence increases with age and varies according to geographical, racial, socioeconomic and ethnic characteristics of the population. In previous seroprevalence studies, HSV-1 antibodies were found in 50–96% of the people tested [12,13,14,15,16]. Prior to acyclovir administration, it is estimated that HSV infection is reactivated in up to 80% of hematopoietic stem cell transplant patients [9,17,18].

Most of the transplant data available are related to hematopoietic stem cell transplantation; however, the potential pathogenicity of HSV during MSC-based therapies is largely unknown. This gap in knowledge is particularly important as MSC therapies (mainly bone marrow, adipose-tissue-derived and Wharton-jelly-derived MSCs) are being used more frequently [19,20,21,22], and donor tissues may contain HSV. Modified HSV is used in melanoma therapy, and aggressive tumors are often linked with tumor-associated cells or tumor stem cells in the majority of cases.

Although it is known that deep non-healing ulcers caused by HSV infection also affect the deeper regions below the skin tissue, it is not known whether the presence of HSV specifically affects stem cells in these tissues. The limited understanding of MSCs’ response to HSV infections can be considered an unknown risk for transplantation outcome, including therapeutic failure [23,24]. Most MSC studies focus on bone marrow [24,25], but there are a considerable number of other clinical uses of adipose-derived MSCs (ADMSCs) [22,26,27,28,29,30]. Adipose tissue itself is not only an important space-filling tissue, but its metabolic activity as an endocrine organ can affect the entire organ system. Furthermore, adipokines produced by ADMSCs can also influence inflammatory processes [31,32,33]. Although terminally differentiated adipocytes have been the focus in studies of viral infections affecting adipose tissue, little is known about ADMSCs [34]. As human ADMSCs can potentially be highly vulnerable cellular targets for viruses, especially for HSV, we investigated how this virus alters MSC viability, phenotype and cytokine secretion pattern.

## 2. Results

Although the cellular responses induced by HSV-1 and HSV-2 are similar, they also show significant differences [35,36,37,38,39]. HSV-1 and HSV-2 have different nucleotide sequences, reactivation rates and transcriptional effects [35,36,37,38,39,40]. There is also a difference between these two viruses in the disease spectrum, the incidence of various clinical manifestations and epidemiological characteristics [35,36,37,38,39]. Most previous studies used the KOS strain of HSV-1, and significant knowledge has been accumulated regarding this strain. We therefore used HSV-1-KOS as a reference strain during our work. As HSV variants can exhibit different genetic and phenotypic features, we also included one wild-type strain each of HSV-1 (HSV-532) and HSV-2 in our study to better understand their effects [41,42]. Based on normalized expression values (z-scores) of the differentially expressed genes (DEGs), HSV1 infections showed a distinct pattern from the control and HSV2 infection (Figure 1A and Appendix A). This observation is explained by the number and similarity of the DEGs. HSV1 KOS infection resulted in the up-regulation of 2062 genes and the down-regulation of 1861 genes. HSV1 532 infection caused the up-regulation of 1035 genes and the down-regulation of 517 genes, and HSV2 infection resulted in the up-regulation of only 80 genes (Figure 1B,D). There was a notable overlap in the DEGs of these three infections. Seventy-five genes were up-regulated in all infections, and a further 752-gene overlap could be seen for the two HSV1 infections. Four-hundred twenty-eight genes were down-regulated in the HSV1 infections (Figure 1C,D).

To identify pathways involved in the altered gene expression, we relied on the KEGG and GO-Gene ontology databases. Pathways upregulated by infection with the HSV-1 KOS strain play roles in the regulation of immune response, tumor necrosis factor-alpha (TNF-α) production, response to viral infection and cell-fate commitment, which are all in the context of cell differentiation (Figure 2A). mRNA processing and other differentiation pathways were also up-regulated by the virus. In the case of the HSV-1 532 strain, immunological pathways were activated, and these pathways were connected to antiviral and immune responses (Figure 2A). The HSV-2 strain induced immune-system processes and responses to external biotic signals. With the help of KEGG (Figure 2B), we were able to identify additional routes. HSV-1 Aries infection caused an up-regulation of the differentiation towards cartilage and bone of mesenchymal stem cells (MSCs), which are involved in the differentiation of many cells and tissues. Different aspects of wound healing, smooth-muscle migration, regulation of epithelial cell death, blood-vessel formation and cellular adhesion pathways were also affected. In subcellular processes, genes regulating different pathways of protein synthesis and NADPH-complex formation were altered. Changes were also detected in the metabolic pathways for alcohol degradation, aerobic respiration and oxidative metabolic stress. With HSV-532 infection, there were significant changes in pathways for ATP metabolism, Golgi-vesicle transport and endoplasmic reticulum formation. The most significant pathways that were down-regulated by the HSV-1 KOS strain are involved in viral protein–cytokine receptor interaction, cytokine and its receptor interaction and COVID-19 infection. The pathways for measles, influenza A and HSV-1 virus infections were also affected (Figure 3A). The pattern of differential regulation was similar for HSV-532, with large changes in pathways specific to hepatitis C and chemokine signaling. Additional pathways were identified via KEGG (Figure 3B): for HSV-1 KOS infection, significant changes were observed for thermogenesis, several aspects of protein synthesis (protein synthesis in ER and o-glycolysis) and proteosome- and lysosome-related gene expression. Pathways specific for prion disease, Parkinson’s disease and Alzheimer’s disease were also affected. After HSV-532 infection, far fewer pathways were affected in ADMSCs and carcinogenesis by reactive oxygen species; oxidative phosphorylation and carbon metabolism processes were affected. The QIAGEN Ingenuity Pathway Analysis (IPA) gave similar results. The typical pathways related to viral infection and viral specific immune response were activated in all treatments (Figure 4A). Changes in ADMSC gene expression were mostly related to cancer and tissue injury pathways (Figure 4B and Appendix A). Table 1 summarizes the top canonical pathways identified using the IPA software based on the significant expression of genes in the in vitro infected and uninfected ADMSCs. The top canonical pathways altered by HSV1-KOS infection included genes involved in mitochondrial dysfunction, oxidative phosphorylation and sirtuin signaling. In addition, interferon signaling, hypercytokinemia/hyperchemokinemia in the pathogenesis of influenza and the activation of IRF (interferon regulatory factors) via cytosolic pattern-recognition receptors were involved in response to HSV1-532 and HSV2 strains.

Some of the signaling pathways were also related to diseases or toxicological pathways, such as for cancer, organismal injury and abnormalities, immunological disease and inflammatory response (Table 2 and Appendix A). Among the most strongly affected genes, *IRF3* and *IFNA2* are members of regulatory pathways related to the replication of herpesviridae, *IRF7* and *SP110* are members of regulatory pathways related to the transactivation of RNA and *IFNG* and Interferon alpha are related to the cell death of epithelial cells Appendix A).

The analysis of gene-expression data has revealed a number of further altered pathways that are associated with cell viability, cell cycle, cell metabolism and cell death. Therefore, we performed additional analyses for the genes involved in this process. Cluster analysis showed that, compared to untreated controls, HSV-2 does not cause as much variation as the HSV1-KOS or HSV-1 532 strains, which cause a more pronounced change in the expression pattern of genes involved in autophagy (Figure 5A). Subcellular morphological lesions characteristic of autophagy were detected in the infected ADMSC culture.

Twenty-four hours after in vitro infection, all HSV strains showed detectable virus particles in both within the cells and in the extracellular space (Figure 5B), as well as in the nucleus (Figure 5(C1,C2)), in the autophagosome (Figure 5(D1,D2)) and in the cytoplasm (Figure 5(E1)), as detected via TEM. Ribosomes were very dense everywhere, which may indicate increased cellular activity and protein production upon HSV infection. There was a slight morphologic difference in the syncytia made by the virus strains. The most transforming effect was exerted by the HSV-1 532 strain, which caused the formation of numerous large multinucleated giant cells containing more than five nuclei per syncytia. Less multinucleated giant cells were formed by the KOS-infected cultures, and the syncytia contained less than five nuclei. In the case of HSV-2 infection, the cytopathic effect was pronounced, and large multinucleated giant cells were produced. LC3 has a central role in autophagy; therefore, fluorescence microscopy was used to confirm the TEM findings by examining the intracellular expression of LC3B in HSV-infected cells. In mock-infected cultures, LCB3 displayed diffuse, cytoplasmic staining patterns, and line-scan fluorescence-intensity studies and 3D surface plots revealed a few peaks that were separated or were more confluent (Figure 6). Cells infected with HSV-1 KOS at multiplicity of infections (MOIs) of 1 and 10, HSV-1 532 at MOIs of 0.01 to 1 and HSV-2 at MOIs of 10 and 100, in contrast, exhibited very bright LC3B staining; the fluorescence intensity profiles were composed of numerous robust peaks, which were separated and cytoplasmic or more confluent and perinuclear. The redistribution as well as the number of autophagic vacuoles at MOIs of 0.1 and 1 increased in ADMSCs (Figure 6). At the protein level, all HSV strains triggered an increase in the ratio of LC3B-II/LC3B-I in ADMSCs at MOI 1, and this observation was confirmed via Western blot analysis. The change in ratio was more dynamic for HSV-1 strains, mainly after 12 h, compared to HSV2 (Figure 7).

Cytokines play a major role in both autophagy and the immunological response to viral infection. Gene-expression data showed that the HSV-1 strains induced significant changes in a subset of cytokines and growth factors compared to the untreated controls (Figure 8A and Appendix A). The effect of HSV-2 was, again, more moderate than HSV-1. To validate our findings at the protein level, altogether, 105 human cytokines from the supernatants of HSV-infected ADMSCs were screened or measured via protein array (Figure 8B and Appendix A). For secreted cytokines, changes in expression pattern detected with cluster analysis showed that all HSV strains caused changes compared to uninfected controls (Figure 8C). Uninfected ADMSCs expressed high levels (at pixel density 28,891 + 1096.02) of angiogenin, also known as potent stimulator of angiogenesis, and of chitinase-3-like protein 1 (CHI3L1), which is expressed and secreted by various cell types, including macrophages, chondrocytes, fibroblast-like synovial cells and vascular smooth-muscle cells. In the control cells, we observed large amounts of cystatin C, which is mainly used as a biomarker for kidney function. Furthermore, large amounts of C-X-C motif chemokine 5 (CXCL5), also known as epithelial-derived neutrophil-activating peptide 78 (ENA78), were observed in the controls and its production following the stimulation of cells with inflammatory cytokines, such as interleukin (IL)-1 or TNF-α [43]. The control cells contained large amounts of endoglin, which is a type I membrane glycoprotein located on cell surfaces and is part of the TGF beta receptor complex, and the growth-regulated oncogene (GRO) alpha or CXCL1 (member of the CXC family), which plays an integral role in the recruitment and activation of neutrophils in response to tissue injury and microbial infection. IL-6 and IL-8 were also present in large amounts in the controls: IL-6 acts as both a pro-inflammatory cytokine and an anti-inflammatory myokine, and IL-8 is a chemokine produced by macrophages and other cell types such as epithelial cells, airway smooth muscle cells [3] and endothelial cells. MCP-1 also could be observed in the control groups; this key chemokine regulates the migration and infiltration of monocytes/macrophages (Figure 8D). RANTES, IP-10 and SDF-1α were not found in the controls, but RANTES was present in varying amounts in all three infected cultures. HSV-KOS-infected ADMSCs produced relatively high levels of chitinase 3-like 1, Dkk-1, endoglin, GRO-α, MCP-1, thrombospondin-1, RANTES and IP-10, whereas small amounts of complement factor D, cystatin C, FGF-19, IGFBP-3, IL-8, MIF and pentraxin-3 were observed.

Fifteen secreted factors were detected in the HSV-532-infected ADMSC culture media; these included angiogenin, chitinase 3-like 1, Dkk-1, DPPIV, ENA-78, endoglin, FGF-19, IGFBP-3, IL-6, MCP-1, MIF, pentraxin-3, RANTES, SDF-1α and thrombospondin-1. When ADMSCs were infected with HSV-2, angiogenin, chitinase 3-like 1, Dkk-1, DPPIV, ENA-78, endoglin, GRO-α, IGFBP-3, IL-6, Il-8, MCP-1, pentraxin-3 and thrombospondin were detectable in the cell supernatants (Figure 8D).

The secreted cytokine pattern was subjected to STRING analysis to assess the biological pathways that are potentially activated by HSV infections in ADMSCs (Figure 9). Cell surface adhesion molecules affect cell–cell interactions, cell–extracellular-matrix (ECM) interactions and, thus, tissue integrity. We determined MSC-specific molecules via flow cytometry. The proportion of CD29-positive cells decreased with HSV-1 532 and HSV-2 infection, but the decrease was not significant due to inter-donor variability. No significant differences were observed for integrin alpha 1, alpha 4 and alpha V after treatments. However, CD54/ICAM and VE-cadherin were significantly altered by the infections (Figure 8E).

## 3. Discussion

MSCs were first described as stromal cells of the bone marrow that exhibited a multipotent differentiation potential and exerted unique immunomodulatory effects [44]. Due to these unique properties, MSCs became one of the most investigated cell types in recent years and have been recognized as perfect candidates to treat some rare and incurable diseases [1]. MSCs have been isolated from many tissues, such as bone marrow, Wharton jelly of the umbilical cord, teeth, placenta and fat tissue. In vitro and in vivo [1,2,3,4,5,6,7] studies demonstrated the robust potential of these cells to differentiate into bone and cartilage tissue, neuronal cells, blood vessels, epithelial cells, cardiac and muscle tissue and more [8,9,10,11]. Furthermore, MSCs have unique immunosuppressive properties which have aroused keen interest in the last few years. As previous reports indicate, MSCs are able to stimulate or suppress immune responses in vitro and in vivo through multiple mechanisms [44,45,46,47,48]. Numerous studies have demonstrated that human MSCs avoid allorecognition, interfere with dendritic cell and T-cell functions and generate a local immunosuppressive microenvironment, which are key steps for tissue homeostasis and regeneration, respectively [46].

Due to their multipotency and immunomodulatory behavior, MSCs are suitable candidates for cell therapy in the field of regenerative medicine [47]. Over the past decade, there has been a significant increase in the number of clinical procedures using visceral and subcutaneous ADMSCs as an alternative to bone-marrow-derived MSCs. The stem-cell microenvironment is of paramount importance in maintaining stem-cell properties, and MSCs are highly sensitive to their environment, especially to inflammatory processes or canonical innate immune responses [24,49,50].

Several viruses are known to infect certain types of MSCs [23,51,52,53,54]. HIV has previously been shown to infect bone marrow MSCs [55]. Viral infection has been shown to impair both the cell viability and proliferative capacity of MSCs, and it leads to alterations in bone and adipose differentiation, which manifests in patients with clinical pathology [51]. Bone marrow MSCs can be infected with herpes viruses, such as HSV-1, HSV-6, HSV-8, varicella-zoster virus (VZV) and cytomegalovirus [25,51,56]. These infections alter the phenotype of MSCs (mainly MHC I and ICAM-1 expression) and also greatly reduce their ability to divide and differentiate [57,58].

Almost all studies confirm that the different activated pattern recognition receptors are highly dependent on the phenotypic and functional appearance of MSCs. However, the descriptions of altered biological functions are reported inconsistently in several studies. Pathogenic patterning specific to certain viruses promotes MSC migration and enhances the immunomodulatory activity of MSCs, while at the same time reducing their viability [24]. In another study, activation via Toll-like receptors (TLRs) determines MSC polarity [59] and, when activated via TLR3 receptors, promotes the generation of MSCs with an anti-inflammatory and immunosuppressive phenotype [59].

Autophagy is a conserved response to cellular stress [60] that is essential for the survival of cells, in particular of cancer cells and stem cells [60,61]. It is known that certain virus-induced pathways (e.g., RIG-I-like receptor signaling) induce autophagy in MSCs, but blocking autophagy does not prevent the activation of cell-death pathways [52]. Our TEM analyses revealed that some HSV-1 and HSV-2 strains elicit strong cytopathic effects and, thereby, may cause severe damage to ADMSCs. The success of ADMSC transplantation can therefore be compromised by HSV reactivation during therapy [56]. The possibility of HSV infection and carriage has also been raised for MSCs of placental origin [59]. MSCs from HIV-infected bone marrow are known to be responsible for the reactivation of the virus [62]. It is not known whether blocking or inducing autophagy would be beneficial in, for example, MSC therapy [63]. The increased autophagic activities of infected cells may affect the viability and immune function of ADMSCs. In line with this finding, we demonstrated here that both HSV-1 and HSV-2 profoundly altered the immunomodulatory function of ADMSCs by triggering the secretion of various cytokines and inflammatory mediators. Elevated levels of IL-6, IL-8, IL-17, CXCL10 and MCP-1 are known to cause cellular senescence in proinflammatory inflammation [64]. Secreted cytokines determine the immunomodulatory capacity of MSCs, but cell surface molecules, as well, are also involved in cell–cell and cell–ECM interactions [65]. CD54 is of paramount importance in MSC immunosuppression [65], and, in our in vitro model, this suppression was altered by the virus, suggesting changed immunological properties of ADMSCs.

During aging, MSCs are exposed to oxidative stress, inflammation or replication depletion. Autophagy can play an important role in these processes [64]. Although there are conflicting data on the precise role of autophagy in inflammation and aging, autophagy is known to be required for the maintenance of cell status for stem-cell progenitors and differentiation potential [60,61,63]. Viral infection may induce inflammation-associated senescence and cause the development of senescence-associated secretory phenotypes in MSCs as well [66]. Thus, our data suggest that the infection of ADMSCs with both HSV-1 and HSV-2 might have a profound effect on the outcome of cell therapies based on ADMSC administration (Figure 9) [24]. In particular, how viral infection can affect MSC function is particularly important when these cells are used to treat organ and tissue damage caused by viral infection [58]. Moreover, some viruses are not able to affect MSC cell differentiation, making MSCs potential therapeutic targets [67].

In many cases, the impact of viral infection on MSC secretions is not known in detail [68]; although, in light of the significant role of MSC-derived exosomes in cancer-metastasis tissue regeneration and the regulation of the microenvironment [69], this information is potentially important. Given the therapeutic implications of MSC-derived exosomes, future research in this area is needed [68,69].

Recently, MSCs have been successfully used in the treatment of SARS-CoV-2-infected patients, which suggests that this coronavirus infection did not cause adverse effects on the cells’ immunomodulatory behavior [70,71].

## 4. Materials and Methods

### 4.1. Cell Cultures

The collection of adipose tissue complied with the guidelines of the Helsinki Declaration and was approved by the National Public Health and Medical Officer Service (NPHMOS) and the National Medical Research Council (16821-6/2017/EÜIG, STEM-01/2017), which follows the EU Member States’ Directive 2004/23/EC on presumed written consent practice for tissue collection. Adipose tissues were removed from patients (age: 43.36 ± 10.07 years; sex: seven female, seven male; see further details in Appendix A) during plastic surgery and processed within 1 h. Briefly, a stromal vascular fraction was produced with a collagenase enzymatic method, and then, cells were plated in 25 cm^2^ cell culture flasks at a density of 2 × 10^5^ living cells/cm^2^ and cultured in low-glucose Dulbecco’s Modified Eagle Medium (Biosera, Nuaille, France), supplemented with 10% fetal calf serum (Biosera, Nuaille, France) and 1% Antibiotic–Antimycotic Solution. At the fifth passage, every cell culture was tested for antigen expression via flow cytometry and in vitro differentiation assays (Appendix A).

### 4.2. HSV Infection of Adipose-Derived Mesenchymal Stem Cellls

The Vero cell line was used for the production and quantification of virus stocks. The KOS strain of HSV-1 (HSV1-KOS), the wild type of HSV-1 (HSV1-532) and the wild-type HSV-2 (HSV2) were propagated at MOI of 0.001 plaque-forming unit per cell in Vero cell cultures for 3 days at 37 °C. The culture fluids of Vero cells infected with either HSV-1 or HSV-2 were harvested, stored at −80 °C and used as the infecting stock of the virus. For the in vitro experiment [42,72,73], ADMSCs were inoculated with HSV-1 or HSV-2 strains at different MOIs and cultured for 24 h [42,72].

### 4.3. RNA-Seq Methods and Data Analysis

To obtain global transcriptome data, high-throughput mRNA sequencing analysis was performed on an Illumina sequencing platform. An Agilent Bioanalyzer with the Eukaryotic Total RNA Nano Kit (Agilent Technologies, Santa Clara, CA, USA) was used to check RNA integrity. RNA samples with DV200 70% were accepted for the library preparation process. mRNA-seq libraries were prepared from total RNA using QuantSeq 3′mRNA-Seq Library Prep Kit for Illumina (FWD) (Lexogen, Vienna, Austria), according to the manufacturer’s protocol. Briefly, first-strand cDNA was generated using an oligo-dT primer linked with an Illumina-compatible sequence. After template RNA removal, the second strand of the cDNA was synthesized using random primers also linked with an Illumina-compatible sequence, and then, double-stranded cDNA was amplified with indexed Illumina-specific adaptors. The sequencing runs were performed on an Illumina NextSeq 500 instrument (Illumina Inc., San Diego, CA, USA) using single-end, 75-cycle sequencing. Reads were trimmed using Trimmomatic [74,75] and aligned to the human genome (GRCh38) with HISAT2. featureCounts [76] was used to create the count table for the gene expression analysis performed in R (version 4.2.0). As a pre-filtering step, genes with low expression values—rows that only had 10 counts across all samples—were removed. To visualize the association between samples, principal component analysis was used with the R package PCAtools (https://github.com/kevinblighe/PCAtools, accessed on 17 April 2023, R package version 2.10.0). Differential expression analysis was performed using DESeq2 [77]. DEGs were defined based on an adjusted *p* value < 0.05 and log_2_-fold change threshold of 0. The heat map visualization of all the DEGs was performed using the R package ComplexHeatmap [78], where the Pearson correlation was used on rows and columns and z-scores were calculated from count data transformed with DESeq2′s variance-stabilizing transformation. EnhancedVolcano package was used to make the volcano plots (https://github.com/kevinblighe/EnhancedVolcano, accessed on 17 April 2023, R package version 1.14.0). For pathway analysis, DEGs ordered by their log-fold changes served as an input for gene-set enrichment analysis using clusterProfiler (version 4.6.2) [79,80,81,82,83,84]. (More information in Appendix A).

### 4.4. Fluorescent Microscopy

For immunohistochemistry studies, cell cultures were fixed in 4% PFA. To visualize endogenous LC3, fixed samples were stained with anti-LC3B antibody (Sigma-Aldrich, St. Louis, MO, USA), and a fluorochrome-conjugated secondary antibody was used to label the cells. Samples were examined under an Olympus IX83 inverted microscope with an Olympus ScanR high-content imaging screening platform (Olympus, Tokyo, Japan) equipped with an Orca2 (Hamamatsu Photonics K.K., Shizuoka, Japan) camera. The images were subjected to line-scan fluorescence-intensity analysis and 3D surface plotting performed with Image J (v.1.53m, 28 September 2021).

### 4.5. Quantification of Virus-Mediated Morphological Changes

One sign of viral infection is the appearance of multinucleated cells in cell culture. The quantification of multinucleated cells was carried out using Olympus CellSens software (version 3.2, build 23706).

### 4.6. Flow Cytometry

A multiparametric analysis of surface-antigen expression was performed via three-color flow cytometry using fluorochrome-conjugated antibodies with isotype matching controls. Samples were measured on a BD FACSAria^™^ Fusion II flow cytometer (BD Biosciences Immunocytometry Systems, Franklin Lakes, NJ, USA) as previously described [85,86]. Data were analyzed using Flowing Software (version 2.5.1, released on 11 April 2013, Cell Imaging Core, Turku Centre for Biotechnology, Finland), and the results were expressed as the means of positive cells (%) ± standard error of mean (SEM).

### 4.7. TEM Analysis

For TEM analysis, cells were fixed with 3% glutaraldehyde solution for 24 h at room temperature and then washed three times for 10 min with 0.1 M sodium cacodylate buffer (pH 7.4) and 7.5% sucrose, followed by post-fixation with 1% OsO_4_ solution for 1 h. The samples were then dehydrated with an ethanol gradient in 70% ethanol (20 min), 96% ethanol (20 min) and 100% ethanol (twice, 20 min) and embedded in Durcupan ACM with uranyl acetate and lead citrate. Sections were examined using a Philips CM-10 TEM (Philips Electronic Instruments, Amsterdam, The Netherlands) at 80 kV.

### 4.8. Detection of Autophagic Flux

To investigate the autophagic flux, bafilomycin A1 (BFLA) was used to prevent the fusion of autophagosomes with lysosomes and block lysosomal degradative activity. The cells were infected either with HSV-1 or HSV-2 at an MOI of 1 and incubated for 24 h. The cultures were then treated with BFLA for an additional 4 h and analyzed for LC3B expression.

### 4.9. Western Blot Analysis of LC3

Infected and treated ADMSCs were collected and washed with ice-cold phosphate-buffered saline and lysed in ice-cold CytoBuster Protein Extraction Reagent (Merck-Millipore, Darmstadt, Germany) containing 1× protease inhibitor cocktail (Thermo Scientific, Waltham, MA, USA). Insoluble cellular debris was removed via centrifugation, and the protein concentration was determined with Bradford reagent (Sigma-Aldrich, St. Louis, MO, USA). The lysates were then mixed with 5× Laemmli loading buffer and boiled for 10 min at 96 °C. Equal amounts of protein (20 µg) were separated in 5% stacking and 15% resolving sodium dodecyl sulfate–polyacrylamide gel. The proteins were transferred onto an Immun-Blot PVDF Membrane (Bio-Rad Laboratories, Inc., Hercules, CA, USA) and subsequently blocked in Tris-buffered saline containing 0.1% Tween-20 (VWR Chemicals Int., France) and 5% non-fat dried milk (AppliChem, Darmstadt, Germany) for 1 h. The membranes were probed with rabbit anti-LC3B antibody (1/1000) (Abcam, Cambridge, UK) and mouse monoclonal anti-GAPDH antibody (1/1000) (Covalab, Villeurbanne, France) in blocking buffer overnight at 4 °C. On the following day, the membrane was incubated with horseradish-peroxidase (HRP)-conjugated secondary antibodies: goat anti-rabbit IgG HRP (Sigma-Aldrich, St. Louis, MO, USA) and rabbit anti-mouse IgG HRP (Bioss, Woburn, MA, USA) in blocking buffer for 1 h at room temperature. Bands were detected with the WesternBright ECL substrate (Advansta Inc., San Jose, CA, USA) and Omega LumG Chemidoc system. Densitometry analysis was performed using Image J software (National Institutes of Health, Bethesda, MD, USA).

### 4.10. Protein Array for the Detection of Cytokines

In this experiment, we aimed to screen the differentially expressed cytokines in ADMSCs infected with different HSV strains. The culture supernatants were collected and pooled from cultures of cells from the donors. All focused protein array analyses were performed according to the manufacturer’s instructions. Positive controls were located in the upper left-hand corner (two spots), lower left-hand corner (two spots) and lower right-hand corner (two spots) of each array kit. Each culture medium was measured using a Proteome Profiler Human XL Cytokine Array kit™ (R&D Systems, Minneapolis, MN, USA), which contains four membranes, each spotted in duplicate with 102 different cytokine antibodies. The pixel density for each spot of the array was determined using ImageJ software [87].

### 4.11. Statistical Analysis

The normality of the distribution of the data was tested using Kolmogorov–Smirnov and Lilliefors tests. Non-normally distributed parameters were transformed logarithmically to correct skewed distributions. R software was used for hierarchical clustering. Each experiment was performed at least three times, and each sample was tested in triplicate. Data are expressed as the mean ± standard deviation or SEM. Statistically significant difference was determined with a one-way analysis of variance (ANOVA) with a post hoc Tukey’s multiple comparisons test when there was more than two groups, whereas analysis between two groups was performed with a paired Student’s *t*-test. A value of *p* < 0.05 was considered significant.

## 5. Conclusions

In conclusion, our results show that in vitro infection with different HSV strains alters the gene expression profile of AD-MSC, stimulates autophagy, and also alters cytokine secretion. Based on our results, it can be assumed that HSV infections can affect the AD-MSCs in the tissue, whose biological processes have changed to reduce wound healing and regenerative processes. can lead to local inflammation. Changes caused by viral infection can affect cell therapies, especially in the case of regenerative medicine and aesthetic interventions, especially where adipose tissue and ADMSc are used (Figure 10). However, some limitations of this study, such as further experimental evidence supporting a reduction in regenerative effect and immunological responsiveness, are needed.

## Figures and Tables

**Figure 1 ijms-24-11989-f001:**
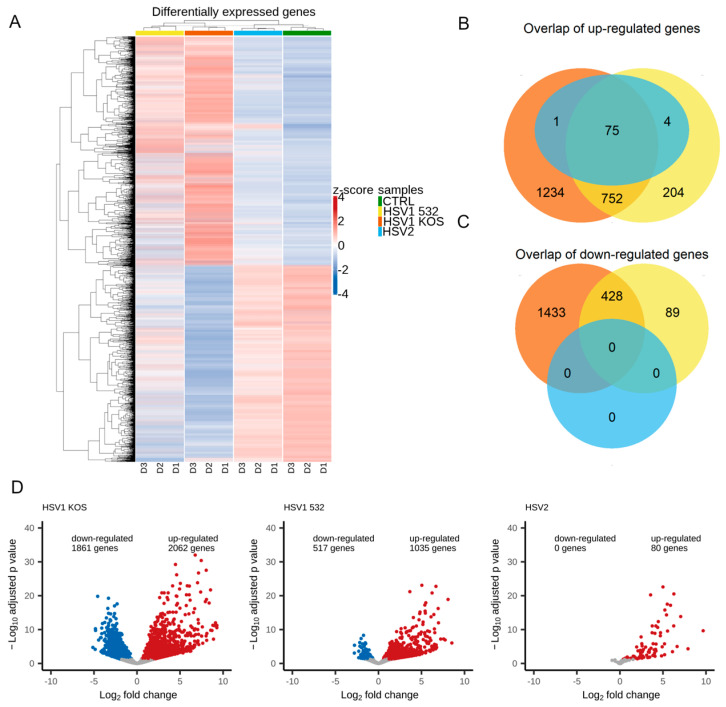
Differentially expressed genes in HSV-infected ADMSCs. (**A**) shows a heatmap of differentially expressed genes after the infection of different virus strains, where D1, D2 and D3 represent the different donors, and the columns represent the samples and the HSV strains are represented by colors. What is most noticeable in this heatmap is that the HSV-2 and CTRL patterns are more similar than HSV1 KOS and HSV1 532. It can be seen that different donors have different patterns. (**B**) We examined the genes that were up-regulated in the HSV1 and HSV2 strains compared to CTRL, and their intersection is visualized by a Venn diagram. (**C**) The numerical distribution of downregulated genes showed that HSV2 did not have any genes that were down-regulated, unlike with the other two HSV-1 strains (**D**). In the volcano plot, the down-regulated genes are marked with blue, the up-regulated genes with red, and those that were not significant based on the adjusted *p*-value with gray. The Log2 fold change value is on the x-axis; it shows how much change there was in the expression of RNAs compared to CTRL. At a log2 fold change of 1, twice as much RNA was transcribed. HSV1 KOS had almost the same number of up- and down-regulated genes, HSV1 532 had far fewer down-regulated genes, and HSV2 only had up-regulated genes. Looking at the KOS dots, several genes were found to be significant.

**Figure 2 ijms-24-11989-f002:**
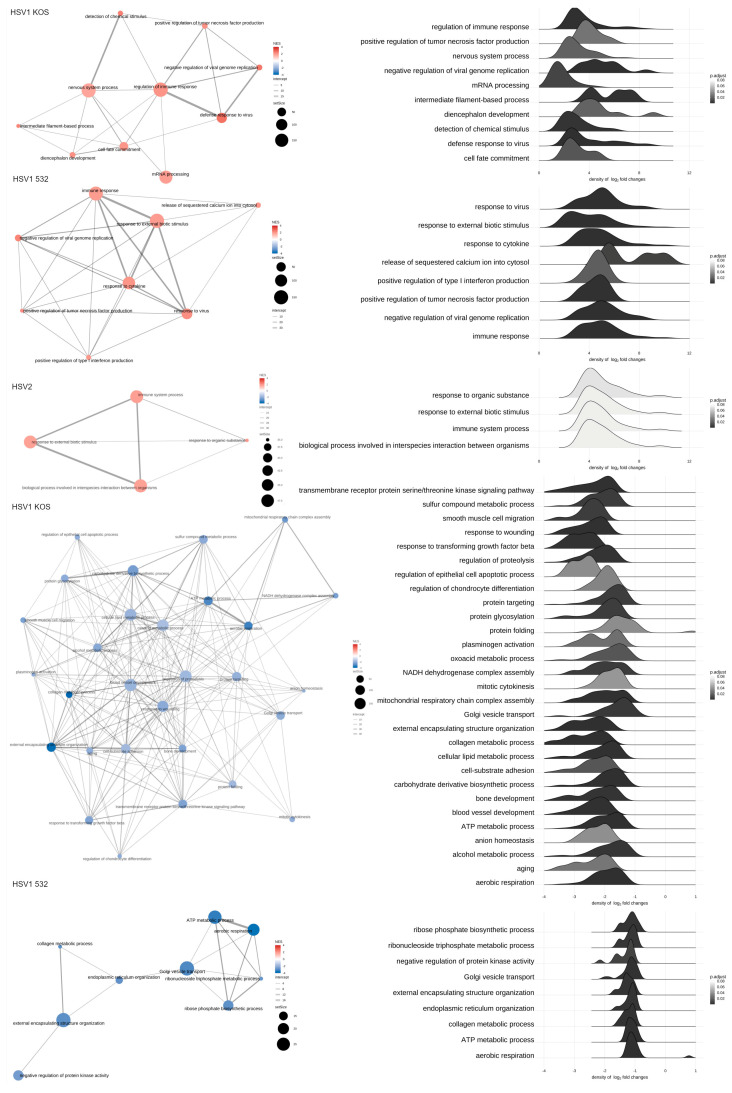
Pathway analysis based on increased gene expression in response to viral infections. GO enrichment and KEGG were used to identify relevant biological pathways based on the increased gene expression patterns.

**Figure 3 ijms-24-11989-f003:**
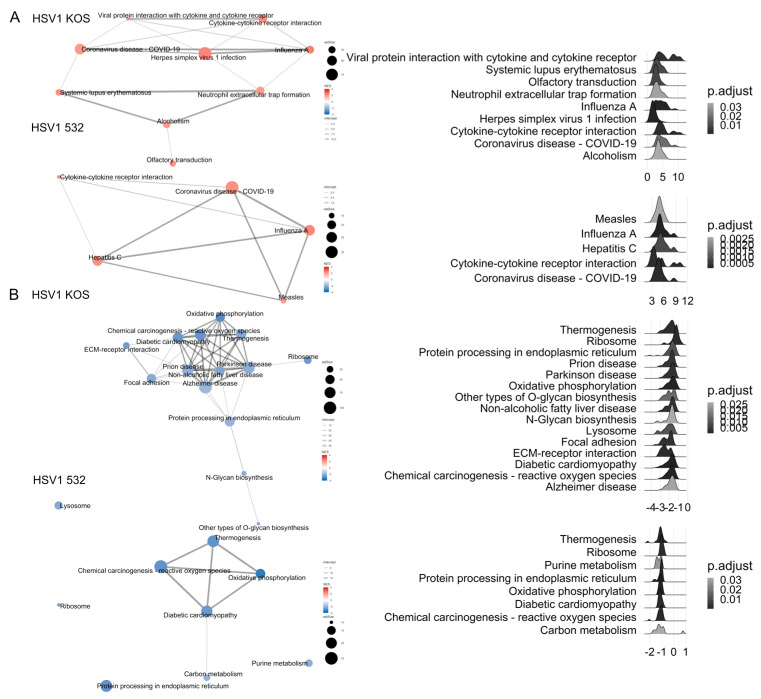
Pathway analysis based on decreased gene expression in response to viral infections. GO enrichment (**A**) and KEGG (**B**) were used to identify relevant biological pathways based on the decreased gene expression patterns.

**Figure 4 ijms-24-11989-f004:**
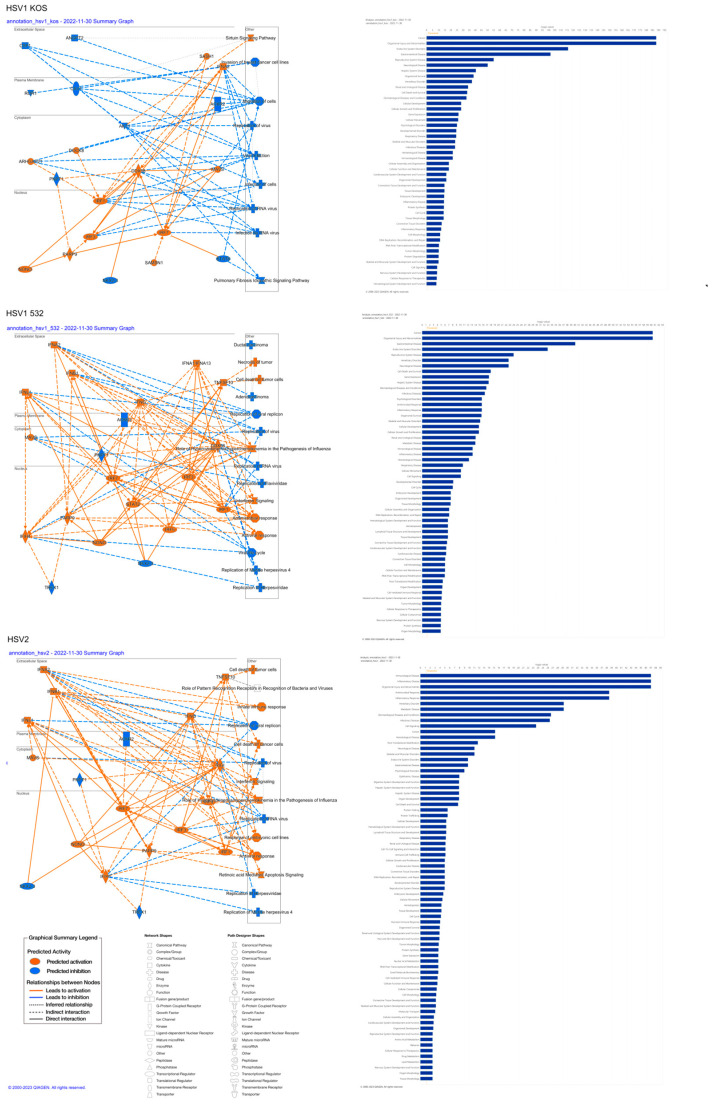
Affected networks and top disease and function pathways of HSV-infected ADMSCs. Graphical summary of IPA analysis on most affected networks in HSV-infected ADMSCs. Most of the networks belong to viral replications and immune responses against viral infection. Detected pathways in disease and function class determined using IPA.

**Figure 5 ijms-24-11989-f005:**
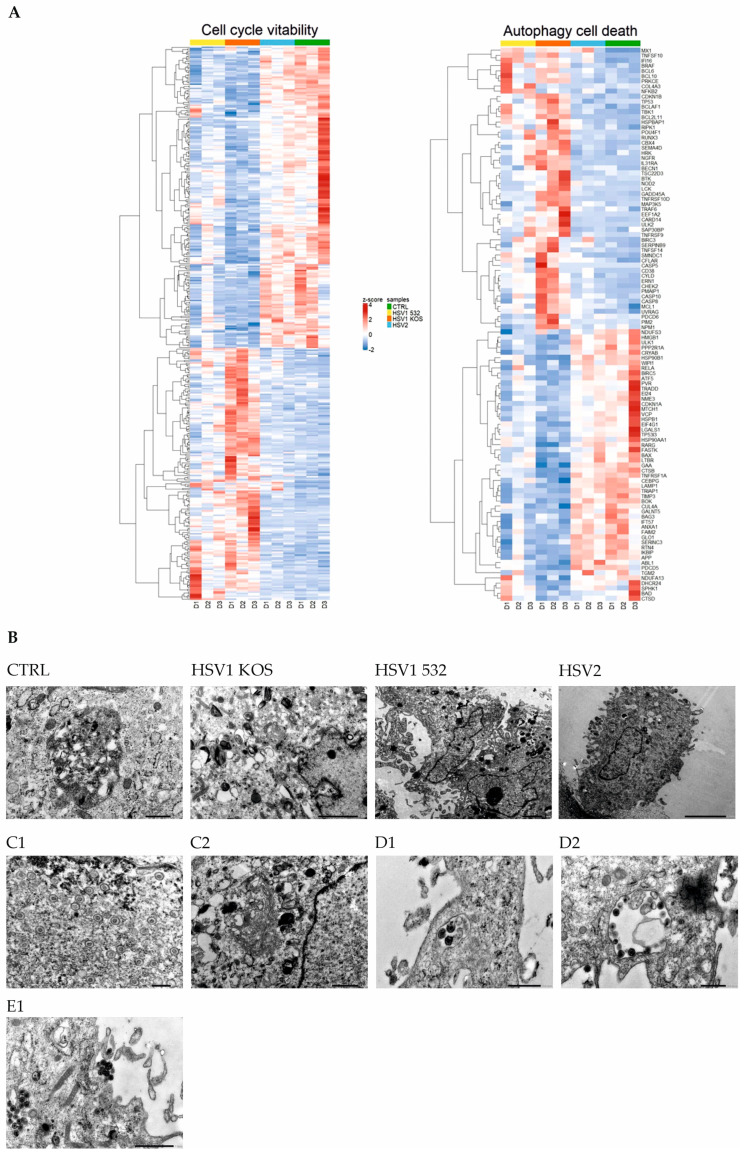
Morphological changes of ADMSCs upon HSV infection. The expression of genes involved in cell cycle and viability, autophagy and cell death showed robust differences between the HSV-infected and uninfected ADMSCs (**A**). HSV replication was examined with the staining of the gD envelope glycoprotein in the infected ADMSCs. Different HSV strains caused the greatest morphological effect at different MOI values, with the appearance of multinucleated cells. HSV virus particles were also detectable via TEM in all viral strains (**B**) in the nucleus (**C1**,**C2**), autophagosomes (**D1**,**D2**) and cytoplasm (**E1**) as well. Results are representative of three independent experiments (TEM M = ×20,000 (**C1**), ×15,000 (**D1**,**D2**), ×12,000 (**E1**)), scale bar 50 μm.

**Figure 6 ijms-24-11989-f006:**
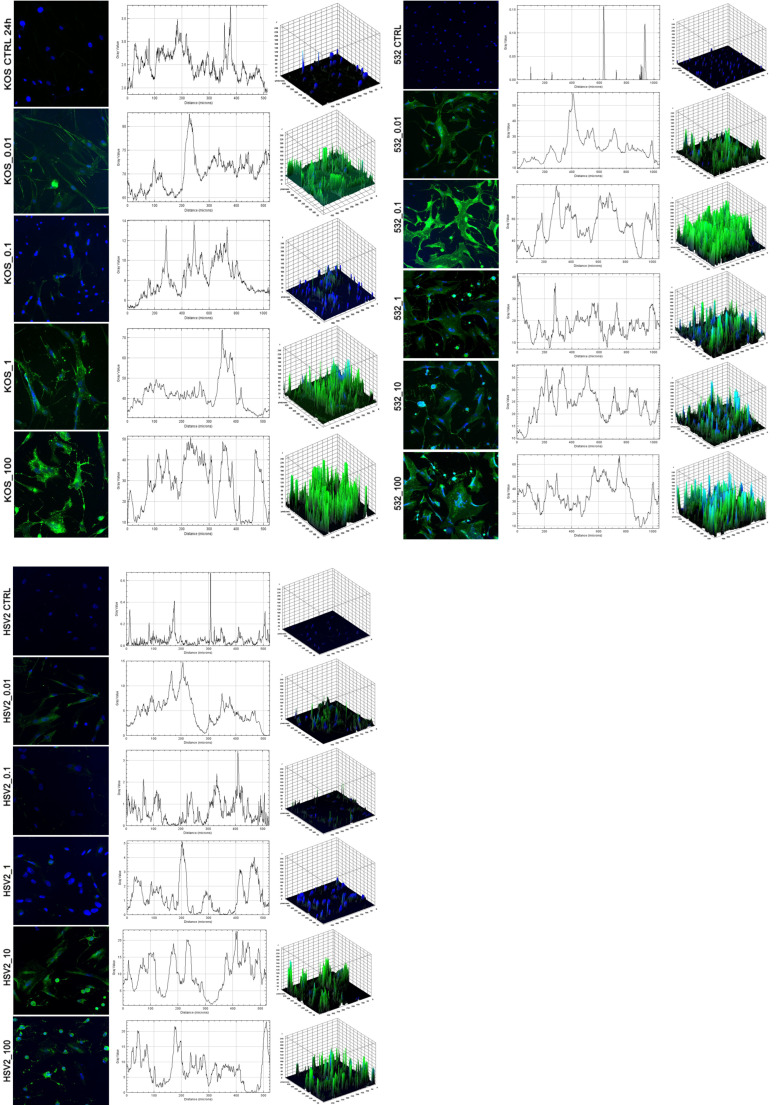
Autophagy in the HSV-infected ADMSCs. The immunofluorescence assay showing the fluorescence intensities of LC3B-positive vacuoles in the HSV-infected ADMSCs. The images were subjected to fluorescence intensity analysis by using the Image J software. The 3D surface plots represent the intensity values of the whole image. The results are representative of three independent experiments.

**Figure 7 ijms-24-11989-f007:**
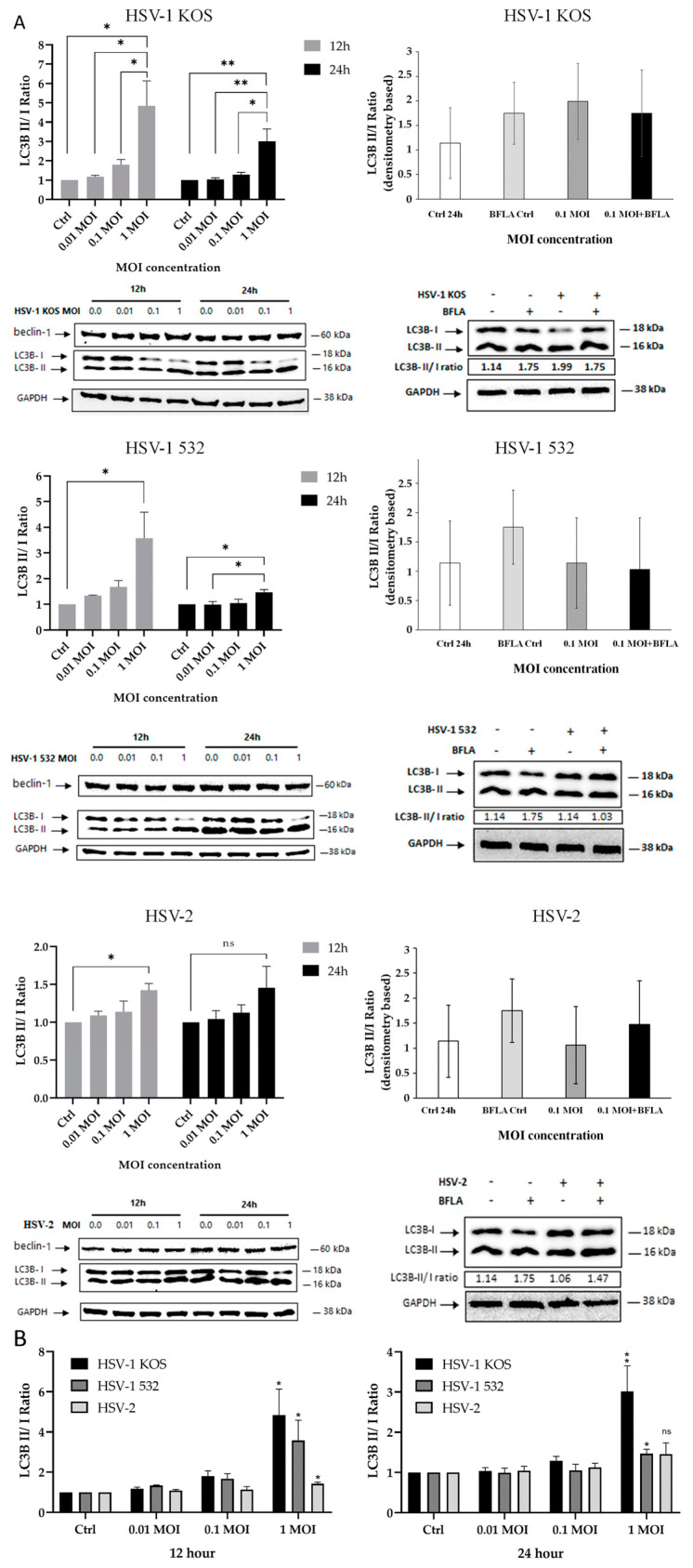
Detection of beclin, LC3B and GAPDH via Western blot. (**A**) Relative densities of LC3B-I, LC3B-II, and GAPDH were measured via densitometric analysis of Western blots. LC3B-I and LC3B-II were normalized to GAPDH, and the LC3B-II to LC3B-I ratio is shown below each lane. (**B**) Comparison of three HSV virus strains in LC3 expression. (N = 3) One-way analysis of variance (ANOVA) was performed, and Tukey’s test assessed differences which were considered significant if *p* ≤ 0.05. (* *p* ≤ 0.05, ** *p* ≤ 0.01).

**Figure 8 ijms-24-11989-f008:**
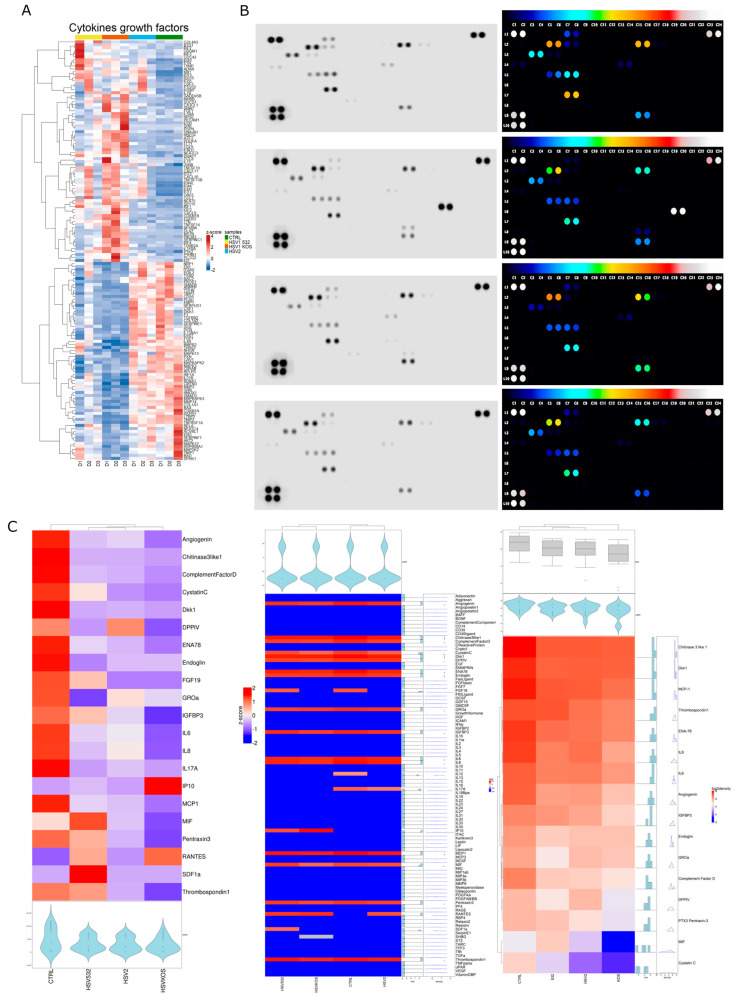
The cytokine secretion and cell surface molecule expression of ADMSCs upon HSV infection. The gene expression pattern revealed to cytokines and growth factors showed distinct differences in the HSV-infected cells (**A**). To examine the cytokine and chemokine production of HSV-infected ADMSC moDCs, the integrated density of cytokines was measured using Human XL Cytokine Array Kit (**B**). The results are visualized in a heatmap where blue and red colors indicate low and high expression, respectively (values represent the integrated density of the plots on the protein array) (**C**). The most significantly changed cytokines after HSV infection. The protein array spots and relative pixel densities of the differentially expressed cytokines with a two-fold change were determined via Image J (**D**). STRING analysis detected the pathways initiated by the secreted cytokines (**E**). The important integrin and CAM molecules of ADMSCs after HSV infection determined via FACS. (Protein array = pooled supernatant from triplicates of three independent donors. Data are presented as relative pixel density).

**Figure 9 ijms-24-11989-f009:**
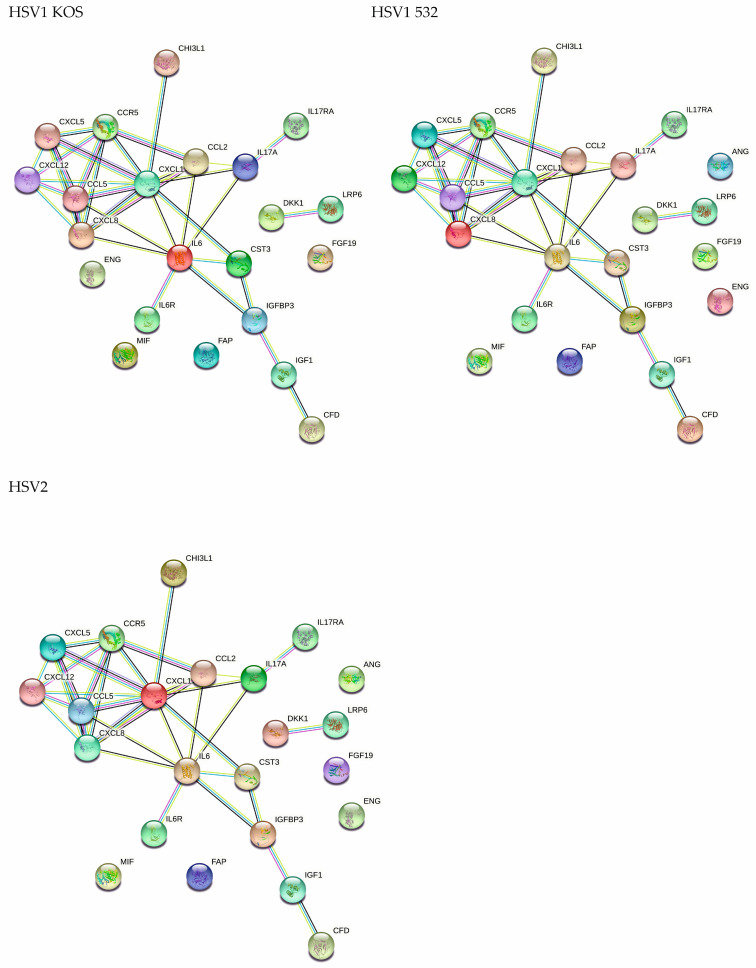
STRING analysis of secreted cytokines by HSV-infected AD-MSCs. Proteins are shown as nodes, and the color of each link defines the type of evidence available for the interaction between two proteins.

**Figure 10 ijms-24-11989-f010:**
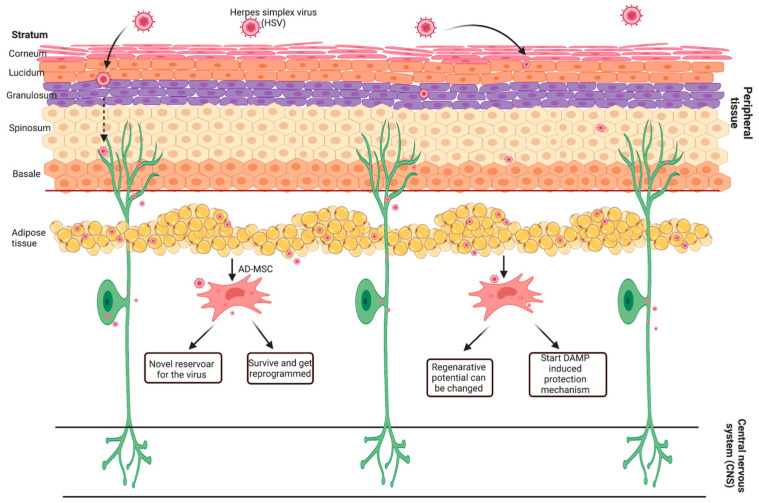
Changes caused by viral infection.

**Table 1 ijms-24-11989-t001:** The most significantly affected canonical pathways found in ADMSCs upon HSV infection.

Top Canonical Pathways			
	Name	*p*-value	Overlap
*HSV1-KOS*	Mitochondrial Dysfunction	6.00 × 10^−12^	36.3% (62/171)
	Oxidative Phosphorylation	2.50 × 10^−10^	39.6% (44/111)
	Pulmonary Fibrosis Idiopathic Signaling Pathway	3.21 × 10^−09^	27.6% (90/326)
	Sirtuin Signaling Pathway	1.83 × 10^−07^	26.6% (78/293)
	Huntington’s Disease Signaling	1.24 × 10^−05^	24.7% (70/283)
*HSV1-532*	Interferon Signaling	3.85 × 10^−06^	30.6% (1/36)
	Role of Hypercytokinemia/Hyperchemokinemia in the Pathogenesis of Influenza	7.84 × 10^−06^	19.8% (17/86)
	Inhibition of ARE-Mediated mRNA Degradation Pathway	2.28 × 10^−05^	14.8% (24/162)
	Activation of IRF via Cytosolic Pattern Recognition Receptors	7.92 × 10^−05^	20.0% (13/65)
	Sirtuin Signaling Pathway	1.01 × 10^−04^	11.6% (34/293)
*HSV-2*	Role of Hypercytokinemia/hyperchemokinemia in the Pathogenesis of Influenza	6.59 × 10^−17^	14.0% 12/86
	Interferon Signaling	2.14 × 10^−11^	19.4% 7/36
	Role of Pattern Recognition Receptors in Recognition of Bacteria and Viruses	4.02 × 10^−08^	5.1% 8/156
	Macrophage Classical Activation Signaling Pathway	1.77 × 10^−07^	4.2% 8/189
	Activation of IRF by Cytosolic Pattern Recognition Receptors	2.19 × 10^−06^	7.7% 5/65
**Top Upstream Regulators**			
Upstream Regulators			
*HSV1-KOS*	Name	*p*-value	Predicted Activation
	beta-estradiol	2.25 × 10^−33^	Inhibited
	MYC	3.35 × 10^−30^	
	TP53	1.97 × 10^−29^	
	TGFB1	1.93 × 10^−27^	Inhibited
	lipopolysaccharide	3.42 × 10^−26^	
*HSV1-532*	NONO	2.11 × 10^−38^	Activated
	CHROMR variant 3	1.67 × 10^−35^	Activated
	IFNL1	2.64 × 10^−33^	Activated
	TREX1	4.88 × 10^−33^	Inhibited
	pyridostatin	2.16 × 10^−32^	Activated
*HSV-2*	NONO	2.61 × 10^−82^	Activated
	IFNL1	1.02 × 10^−78^	Activated
	pyridostatin	9.69 × 10^−73^	Activated
	IFNA2	1.19 × 10^−68^	Activated
	TREX1	5.75 × 10^−67^	Inhibited
**Causal Network**			
*HSV1-KOS*	beta-estradiol	1.17 × 10^−48^	Inhibited
	CCNT1	6.16 × 10^−44^	Inhibited
	BMS-690514	7.17 × 10^−44^	
	RPL11	1.58 × 10^−43^	
	PA2G4	3.68 × 10^−43^	
*HSV1-532*	NONO	7.25 × 10^−39^	Activated
	CHROMR variant 3	1.67 × 10^−35^	Activated
	IFNL1	1.65 × 10^−34^	Activated
	pyridostatin	2.16 × 10^−32^	Activated
	STAG2	6.77 × 10^−31^	Inhibited
*HSV-2*	NONO	1.17 × 10^−82^	Activated
	IFNL1	8.99 × 10^−80^	Activated
	pyridostatin	9.69 × 10^−73^	Activated
	IFNA2	8.54 × 10^−69^	Activated
	STAG2	3.63 × 10^−66^	Inhibited

**Table 2 ijms-24-11989-t002:** The most significantly affected diseases and bio functions found in ADMSCs upon HSV infection.

Top Diseases and Bio Functions			
Diases and Disorders			
*HSV-1 KOS*	Name	*p*-value range	# Molecules
	Cancer	8.05 × 10^−09^–5.38 × 10^−189^	3425
	Organismal Injury and Abnormalities	8.47 × 10^−09^–5.38 × 10^−189^	3482
	Endocrine System Disorders	4.28 × 10^−09^–9.25 × 10^−117^	2942
	Gastrointestinal Disease	6.99 × 10^−09^–2.30 × 10^−102^	3038
	Reproductive System Disease	3.98 × 10^−09^–8.17 × 10^−56^	2442
*HSV-1 532*	Cancer	1.92 × 10^−05^–4.01 × 10^−61^	1323
	Organismal Injury and Abnormalities	1.94 × 10^−05^–4.01 × 10^−61^	1347
	Gastrointestinal Disease	1.82 × 10^−05^–7.75 × 10^−41^	1181
	Endocrine System Disorders	1.08 × 10^−05^–1.28 × 10^−33^	1125
*HSV-2*	Immunological Disease	3.23 × 10^−03^–8.71 × 10^−48^	59
	Inflammatory Disease	3.23 × 10^−03^–8.71 × 10^−48^	51
	Organismal Injury and Abnormalities	3.23 × 10^−03^–8.71 × 10^−48^	75
	Antimicrobial Response	3.23 × 10^−03^–2.90 × 10^−39^	33
	Inflammatory Response	3.23 × 10^−03^–2.90 × 10^−39^	53
**Molecular and Cellular Functions**			
*HSV-1 KOS*	Name	*p*-value range	# Molecules
	Cell Death and Survival	3.72 × 10^−09^–4.41 × 10^−34^	1274
	Cellular Development	6.54 × 10^−09^–3.79 × 10^−29^	1045
	Cellular Growth and Proliferation	6.54 × 10^−09^–3.79 × 10^−29^	953
	Gene Expression	9.96 × 10^−11^–5.57 × 10^−27^	788
	Cellular Movement	6.39 × 10^−09^–9.76 × 10^−27^	939
*HSV-1- 532*	Cell Death and Survival	1.62 × 10^−05^–1.05 × 10^−18^	517
	Gene Expression	1.29 × 10^−05^–3.07 × 10^−18^	356
	Cellular Development	1.79 × 10^−05^–1.30 × 10^−15^	448
	Cellular Growth and Proliferation	1.79 × 10^−05^–1.30 × 10^−15^	418
*HSV-2*	Cell Signaling	2.14 × 10^−04^–2.37 × 10^−24^	17
	Post-Translational Modification	2.23 × 10^−03^–1.84 × 10-^12^	10
	Cell Death and Survival	3.23 × 10^−03^–1.94 × 10^−08^	36
	Protein Folding	2.68 × 10^−06^–2.68 × 10^−06^	3
	Protein Trafficking	3.23 × 10^−03^–2.68 × 10^−06^	8
**Physiological System Development and Function**			
*HSV-1 KOS*	Name	*p*-value range	# Molecules
	Organismal Survival	7.18 × 10^−09^–2.40 × 10^−39^	1011
	Cardiovascular System Development and Function	8.04 × 10^−09^–5.20 × 10^−17^	410
	Organismal Development	3.02 × 10^−09^–5.20 × 10^−17^	1083
	Connective Tissue Development and Function	4.63 × 10^−09^–1.78 × 10^−15^	594
	Tissue Development	4.63 × 10^−09^–1.78 × 10^−15^	757
*HSV-1- 532*	Organismal Survival	1.17 × 10^−05^–2.74 × 10^−16^	395
	Embryonic Development	1.58 × 10^−05^–3.05 × 10^−08^	221
	Organismal Development	1.76 × 10^−05^–3.05 × 10^−08^	404
	Tissue Morphology	1.82 × 10^−05^–3.40 × 10^−08^	289
	Hematological System Development and Function	1.58 × 10^−05^–1.70 × 10^−07^	212
*HSV-2*	Digestive System Development and Function	3.23 × 10^−03^–1.25 × 10^−08^	11
	Hepatic System Development and Function	3.23 × 10^−03^–1.25 × 10^−08^	11
	Organ Development	3.23 × 10^−03^–1.25 × 10^−08^	18
	Hematological System Development and Function	3.23 × 10^−03^–5.14 × 10^−06^	28
	Lymphoid Tissue Structure and Development	3.23 × 10^−03^–5.14 × 10^−06^	24

## Data Availability

All data generated and analyzed during this study are included in this manuscript (and its Appendix A).

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
