# Peer review of "Herpes Simplex Virus Infection Alters the Immunological Properties of Adipose-Tissue-Derived Mesenchymal-Stem Cells"

_ijms, 2023, doi:10.3390/ijms241511989_

Round 1
Reviewer 1 Report
The authors present the manuscript Herpes Simplex Virus Infection Alters the Immunological Properties of Adipose Tissue De-rived Mesenchymal Stem like Cells
In this study the authors describe the response of adipose-derived MSC cells (ADMSC) to HSV infection in vitro. The authors identified 16 significantly altered cytokines involved in tissue regeneration and inflammation .
However it is not clear the authors have shown an effect associated with autophagy.
Additionally there appears to be little to no verification of cytokine levels as determined by ELISA. (Instead of dot blots presented in Figure 8)
The authors should carefully perform additional functional characterization of HSV infected cells to asses the quality of transcriptomic data analysis presented in figures 1-4
In figure 5 could the authors provide scale bars to indicate the level of magnification
In addition to figure 7. Could the authors show any data on apoptosis or cell proliferation.
THe manuscript is decently written but should be edited for grammatical errors.
Please kindly resubmit the manuscript after addressing these concerns.
Moderate editing of English language
Author Response
Thank you for the well-detailed and thorough critique of the manuscript, making it more comprehensible and of higher quality.
Please find our detailed response to the raised questions in the attached pdf.

Reviewer 2 Report
Thank you for the opportunity to review the very interesting, and clinically relevant, article looking at HSV infection in AD-MSCs. I would like to offer a few suggestions:
Introduction:
Line 53-55: provide ref.s
Ln 59-61: very confusing sentence. Need to be re-phrased.
Ln 66: you have used relatively narrow evidence for your prevalence data, focusing on specific locations. Are there any systematic reviews available?
Ln 67: Confusing. Is acyclovir a relatively new treatment and all px are now given it, so that the 80% re-activation no longer occurs?
Ln 74-75: Provide ref.s confirming AD-MSC are one of the 3 most used MSCs in cell based-therapies. Regarding donor cells being contaminated with HSV; are the cells screened prior to patient use?
Ln 76: reference?
Ln 83: again, no reference supporting the frequent use of AD-MSCs.
Ln 83-85: references?
Results:
Please explain the different strains of HSV you used, and why.
Figure 7. Could all 3 strains of HSV have been plotted on the same graph, to allow easier comparison?
Methods:
Ln 318 – age range unclear
I could not access suppl. table 1 and 2. and would like to see how you characterized your AD-MSCs.
Section 4.2: did you use a control of Vero cells only for your analysis?
4.11: What post-hoc test did you use for ANOVA?
Conclusion:
Much of this could have been in the introduction. Needs re-writing to be more specific to discussion of your data/results.
Overall:
Why did you use adipose-derived stem cells? Are these the stem cells most likely to come into contact with HSV or to be used in cell therapies? This is not clear and references supporting their use were not well provided.
Some references are missing when you are discussing specific uses/functions of your cells/HSV.
The manuscript would benefit from another proof-read, as there were some grammatical errors, and sentences where it was not clear exactly what you were trying to say.
Author Response

(The authors gave the same response as above.)

Round 2
Reviewer 1 Report
The authors addressed reviewer concerns
The authors addressed reviewer concerns